# Consumer engagement in health care policy, research and services: A systematic review and meta-analysis of methods and effects

Louise K. Wiles[1,2,3], Debra Kay[4], Julie A. Luker[1], Anthea Worley[1], Jane Austin[5], Allan Ball[6], Alan Bevan[4], Michael Cousins[7], Sarah Dalton[8,9], Ellie Hodges[10], Lidia Horvat[11], Ellen Kerrins[12], Julie Marker[4], Michele McKinnon[13], Penelope McMillan[4], Maria Alejandra Pinero de Plaza[4,14,15], Judy Smith[16], David Yeung[3,17,18], Susan L. Hillier[1]*

1 Allied Health and Human Performance, University of South Australia, Adelaide, South Australia, Australia, 2 Centre for Healthcare Resilience and Implementation Science, Australian Institute of Health Innovation, Macquarie University, North Ryde, New South Wales, Australia, 3 South Australian Health and Medical Research Institute (SAHMRI), Adelaide, South Australia, Australia, 4 Health Consumer Advocate/ Representative, Adelaide, South Australia, Australia, 5 Health Performance Council, Adelaide, South Australia, Australia, 6 National Disability Insurance Agency, Adelaide, South Australia, Australia, 7 Carers SA, Seaton, South Australia, Australia, 8 The Children's Hospital at Westmead, Westmead, New South Wales, Australia, 9 Agency for Clinical Innovation, St Leonards, New South Wales, Australia, 10 Lived Experience Representative, Adelaide, South Australia, Australia, 11 Safer Care Victoria, Melbourne, Victoria, Australia, 12 SAHMRI Community Advisory Group, Adelaide, South Australia, Australia, 13 SA Health, Adelaide, South Australia, Australia, 14 Caring Futures Institute, College of Nursing and Health Sciences, Flinders University, Bedford Park, South Australia, Australia, 15 National Health and Medical Research Council Transdisciplinary Centre of Research Excellence in Frailty Research to Achieve Healthy Ageing, Adelaide, South Australia, Australia, 16 Royal District Nursing Service, Keswick, South Australia, Australia, 17 Royal Adelaide Hospital, Adelaide, South Australia, Australia, 18 SA Pathology, Adelaide, South Australia, Australia

☯ These authors contributed equally to this work.
‡ JA, AB, AB, MC, SD, EH, LH, EK, JM, MM, PM, MAPP, JS and DY also contributed equally to this work.
* susan.hillier@unisa.edu.au

**Data Availability Statement:** All relevant data are within the paper and its Supporting Information files.

## Abstract

To assess the effects of consumer engagement in health care policy, research and services. We updated a review published in 2006 and 2009 and revised the previous search strategies for key databases (The Cochrane Central Register of Controlled Trials; MEDLINE; EMBASE; PsycINFO; CINAHL; Web of Science) up to February 2020. Selection criteria included randomised controlled trials assessing consumer engagement in developing health care policy, research, or health services. The International Association for Public Participation, Spectrum of Public Participation was used to identify, describe, compare and analyse consumer engagement. Outcome measures were effects on people; effects on the policy/ research/health care services; or process outcomes. We included 23 randomised controlled trials with a moderate or high risk of bias, involving 136,265 participants. Most consumer engagement strategies adopted a **consultative** approach during the **development** phase of interventions, targeted to **health services**. Based on four large cluster-randomised controlled trials, there is evidence that consumer engagement in the development and delivery of health services to enhance the care of pregnant women results in a reduction in neonatal, but not maternal, mortality. From other trials, there is evidence that involving consumers in

**Funding:** The author(s) received no specific funding for this work.

**Competing interests:** The authors have declared that no competing interests exist.

developing patient information material results in material that is more relevant, readable and understandable for patients, and can improve knowledge. Mixed effects are reported of consumer-engagement on the development and/or implementation of health professional training. There is some evidence that using consumer interviewers instead of staff in satisfaction surveys can have a small influence on the results. There is some evidence that consumers may have a role in identifying a broader range of health care priorities that are complementary to those from professionals. There is some evidence that consumer engagement in monitoring and evaluating health services may impact perceptions of patient safety or quality of life. There is growing evidence from randomised controlled trials of the effects of consumer engagement on the relevance and positive outcomes of health policy, research and services. Health care consumers, providers, researchers and funders should continue to employ evidence-informed consumer engagement in their jurisdictions, with embedded evaluation.

**Systematic review registration:** PROSPERO CRD42018102595.

## Introduction

Engaging consumers in health care decisions is widely recognised as being important in health care policy, research and services. Consumer participation can be viewed as a goal in itself, by encouraging participative democracy, public accountability and transparency. Consumers may offer different and complementary perspectives and priorities to those of professionals; furthermore, they may not have the same conflicts of interest and loyalties as professionals. The concept of consumer engagement is founded on the principle that health care policy, research and health services are in the public interest [1] and people have the right to be engaged with and contribute to decisions which will affect them [1, 2]. In their review, Degeling et al [3] found the purpose for involving consumers in health policy processes is to capture the plethora of community perspectives, to enable consumer responsibility, and to examine acceptability of approaches to generate evidence for policymaking [3]. In health research, evidence supports the notion that consumer engagement leads to research of greater quality and clinical relevance [4] and application of findings [5]. There is also some evidence that input from consumers in planning health care can lead to more accessible and acceptable health services [6]. That said, there is a lack of contemporary research that reliably or systematically investigates whether consumer engagement achieves these intended benefits and if so, which methods of consumer engagement are most effective, and how these effects might be measured.

Despite the development of policy to support consumer engagement, there is evidence of widespread national and international variation in the extent to and the manner in which consumers are engaged. For example, membership of peak grant committees has been found to be dominated by academics and clinicians in over 70% of eleven nationally-based research funding organisations recently surveyed, and only one organisation provided public access to full protocols for completed or ongoing research [7]. Furthermore, and notwithstanding the availability of well-established standards for consumer (stakeholder) engagement in creating clinical practice guidelines [8–10], there are considerable inconsistencies and gaps in practice [11–13]. Conversely, several consumer engagement strategies have led to tangible improvements across a range of metrics. The Guidelines International Network sought to operationalise their

published standards for guideline development [9] as a toolkit [14] employing a mixed approach of literature reviews and stakeholder panels, they specifically focused on ways to meaningfully involve patients and members of the public in developing guidelines. In addition, a number of initiatives have been implemented by organisations such as *Planetree International* (e.g. consulting and training programs), The King's Fund (e.g. involving patients and carers in research, conference planning and collaborative leadership) [15], and the UK *National Health Service* project aimed at exploring how patients and carers can act as leaders, provide feedback and improve the experience of healthcare (e.g. 'patients as leaders' which resulted in the generation of building blocks for success and identification of key roles across systems) [16].

Many health professionals, consumers and organisations are calling for hard evidence and robust evaluations of many factors around consumer engagement including impact, how it leads to benefits, the best method for translation and implementation and even how it should be conceptualized [17–22]. Not all of these gaps in knowledge and application can be addressed by this review. However, we can aim to identify good practice to support and advise uptake in areas that are currently active, as well as provide evidence to support engagement in settings where it is not as common. A review was first published on this topic in 2006 and updated in 2009 [23]. Since 2009 there have been numerous developments regarding consumer engagement (sometimes termed 'patient and public involvement') in health care services, policy and research. This has been accompanied by considerable growth in people and organisations undertaking consumer engagement [24, 25], shifts in terminology [26], and new conceptual models and frameworks to explore, explain and evaluate consumer engagement in health [27–30]. In our review which updates the 2009 review [23], we aim to report on: (a) the methods of consumer engagement strategies used in societal decisions on health care policy, research and services (according to the IAP2 Public Participation Spectrum); and, (b) their effects on the people involved in the engagement, on the research/policy/health care service, as well as process outcomes.

The primary objective of the review was to assess the effects of consumer engagement on health care policy, research and services [23]. Secondary objectives were to explore whether differences between studies might explain any differences between the effects [23]. We were specifically interested in differences in the:

- methods (levels) of consumer engagement (e.g. fact sheets, focus groups, patient advisory committees representing the levels of inform, consult, involve, collaborate or empower) [23];

- stages (i.e. development, implementation, monitoring, evaluation) of health care policy, research and services in which consumers are engaged [23]; and

- characteristics of consumer or professional participants (e.g. background, experience or training in consumer engagement) [23].

## Methods

Human Research Ethics Committee approval was granted from the University of South Australia (protocol number 0000036486) and La Trobe University (approval number S17-013). Written and oral consent was obtained.

### Patient and public involvement statement

A new author team was formed to undertake an update on the 2009 review [23] that included researchers and a consumer representative. In addition, we elected to partner with a

stakeholder advisory group (including ten consumers) to enhance the relevance and currency of the review to potential readers and users. Further details of the stakeholder group members and the engagement process are included in S1 Appendix. Given the significant developments since the last review, the author team as advised by the stakeholder group, deemed the publication of a new review protocol and results to be relevant and necessary [31]. At key stages of the systematic review process, stakeholders were invited to provide perspectives and feedback which were used to: craft and refine the research question(s) and definitions for the population, intervention, comparator/control, outcome [PICO] criteria; contextualize initial analyses of results from included studies; and ensure the appropriateness of interpretations from the study findings in the draft final review report.

## Criteria for considering studies for this review

Our review protocol was published *a priori* in PROSPERO [31]; for further details of our methods please refer to this listing. For our working definitions of consumer engagement, health care consumer, health care professional/researcher/policy-maker, health care policy, health care services, health care research and further descriptors of potential outcomes, S2 Appendix.

## Our definition of health care consumer

We used the following definition of health care consumer: patients and potential patients, carers, and people who use health care services. Collectively, 'consumers' and 'community members' may be referred to as 'the public' [32]. However, given the variations in terminology within different contexts, we included any of the following terms for health care consumers: patients; unpaid carers (current or former); parents/ guardians/family; users and potential users of health care services; people with lived experience; peer workforce; people with disability; members of the public who are the potential recipients of health promotion/public health programmes; groups asking for research because they believe they have been exposed to potentially harmful circumstances, products or services; groups asking for research because they believe they have been denied products or services from which they believe they could have benefited; and organisations that represent service users and carers [23]. Depending on the context, they could be described with any of the following terms: 'lay', 'service user', 'survivor', 'patient and public involvement' or 'member of the general public' [23]. We acknowledge that our broad definition means that every individual would be included as a health care consumer. As such, we focused on the role played; that is, only included participants as health care consumers when they had been engaged in health care services, policy or research with the identifiable purpose of bringing a consumer perspective.

## Our definition of health care professional, researcher or policy maker

We used the following definition of a health care professional, researcher or policy maker: people who are employed in health care services, research institutions or government health departments or related agencies as health care professionals (in any professional discipline), health care service managers, researchers, and policy-makers who participate in the included study according to one (or more) of these roles [23].

## Our definition of 'consumer engagement'

We used the following definition of consumer engagement: "an informed dialogue between an organisation and consumers, carers and the community which encourages participants to

share ideas or options and undertake collaborative decision making, sometimes as partners" [33].

As discussed, we have chosen to use the IAP2 Public Participation Spectrum to help us determine what should be included as a consumer engagement study [34]. Examples of methods of engagement (for each IAP2 participation level) include: mass media and fact sheets (inform), focus groups and patient surveys (consult), patient advisory committees and Delphi processes (collaborate), and citizen panels or consumer managed projects and services (empower) [35]. As such, we included studies that described a consumer engagement activity that met the criteria for consult, involve, collaborate or empower (i.e. all items with the exception of 'inform' as it does not fit with our definition of consumer engagement being a two-way 'informed dialogue'). This is also consistent with the way in which the UK's NIHR INVOLVE [26] defines what they term 'public involvement' in research, clarifying it does not include 'researchers raising awareness of research, sharing knowledge or engaging and creating a dialogue with the public'.

Given the different terminology used to describe consumer engagement, we accepted any terminology used in the studies (i.e. participation, involvement, co-production, co-design), as long as the description about what was involved met our criteria for engagement.

## Our definition of health care policy

We defined health care policy as "decisions, plans, and actions that are undertaken to achieve specific health care goals within a society. An explicit health policy can achieve several things: it defines a vision for the future which in turn helps to establish targets and points of reference for the short and medium term. It outlines priorities and the expected roles of different groups; and it builds consensus and informs people" [36]. Common outputs of health care policy include standards, practice guidelines or position statements. We included studies in health care policy undertaken by any health care organisations, e.g. national, state and local governments, non-government organisations, health care services, private organisations or consumer groups, and at any stage of the policy-making cycle (commonly described as agenda setting, formulation, adoption, implementation, and evaluation).

## Our definition of health care services

We defined health care as "services provided to individuals or communities by health service providers for the purpose of promoting, maintaining, monitoring or restoring health" [37]. Health care services providing direct care to patients in primary, secondary or tertiary settings were included in this review. Within this context, consumers might be involved in activities like health care service governance, health care service redesign, developing patient information for informed decision-making/consent, among others. An important distinction is that we did not include studies where the aim was to engage consumers in their own individual care, but rather they are involved in broader activities of the health care service.

## Our definition of health care research

We defined health care research as clinical research, epidemiological research and health care services research (investigating need, demand, supply, use, and outcome of health care services) [38]. This also included public health and health promotion research. Within this context, consumers might be involved in research funding decisions, setting research priorities, and planning, undertaking or disseminating research, among others. This does not include studies where the only role consumers have is as a participant of the study.

### Types of studies

Randomised controlled trials (RCTs), cluster-RCTs and quasi-RCTs (as defined by the Cochrane Collaboration Handbook) [39].

### Types of participants

We included studies investigating the effects of consumer engagement on health care services, policy or research. There are two layers of participants in this review: (1) 'Engagement participants' who are involved in the engagement process and (2) 'Intended recipient participants' of the health care policy, research or services that have been developed, implemented, monitored, and/or evaluated using the consumer engagement strategy. 'Engagement participants' are health care consumers and professionals (meaning health care practitioners, researchers or policymakers).

### Types of interventions

We defined consumer engagement as "an informed dialogue between an organisation and consumers, carers and the community which encourages participants to share ideas or options and undertake collaborative decision making, sometimes as partners" [33], and used the IAP2 Public Participation Spectrum to help us determine what should be included as a consumer engagement study [28]. We included studies which compared consumer engagement to no consumer engagement or compared one method of consumer engagement to another method of consumer engagement, in the context of health care policy, research or services, where that engagement met the criteria for consult, involve, collaborate or empower (i.e. all items with the exception of 'inform' as it does not fit with our definition of consumer engagement being a two-way 'informed dialogue').

### Types of outcome measures

To be included, a trial must have had a quantitative measure, based on the following three broad outcome categories to describe the range of effects: effects on people; effects on health care policy/research/ services; and process outcomes (S2 Appendix).

### Search methods for identification of studies

We revised previous search strategies (S3 Appendix) and searched the following databases without language restriction: The Cochrane Central Register of Controlled Trials (CENTRAL, The Cochrane Library, February 2020); MEDLINE (OvidSP) (2009 to February 2020); EMBASE (OvidSP) (2009 to February 2020); PsycINFO (OvidSP) (2009 to February 2020); CINAHL (EbscoNet) (2009 to February 2020); Web of Science (2009 to February 2020).

We searched the following additional sources and places for published and unpublished studies: websites of relevant organisations; clinical trials registries; Google Scholar; reference lists of included studies; and citation tracking of included studies. In addition, we liaised with our stakeholder group and contacted experts in the field directly. We also promoted our review on Twitter and Facebook, inviting people to send us studies. All revised and updated search strategies are available from the principal author.

### Data collection and analysis

Two authors (two of LW, AW, JL, SH) independently screened all titles and abstracts identified from searches to determine which met the inclusion criteria, with the assistance of Covidence systematic review software [Covidence systematic review software, Veritas Health Innovation,

Melbourne, Australia]. We retrieved in full text any papers identified as potentially relevant. Two review authors independently screened full-text articles for inclusion or exclusion, with discrepancies resolved by discussion and consulting a third author, if necessary, to reach consensus. A pilot screening of 100 papers was conducted to ensure all criteria were being applied consistently across studies by the four authors. All potentially relevant papers excluded from the review at this stage are listed as excluded studies, with reasons available from the authors. We also noted citation details and any available information about ongoing studies and collated and reported details of duplicate publications, so that each study (rather than each report) is the unit of interest in the review. Studies with more than one reference were identified by the year the study was conducted or completed. We report the screening and selection process in an adapted PRISMA flow chart (Fig 1) and checklist (S4 Appendix) [40].

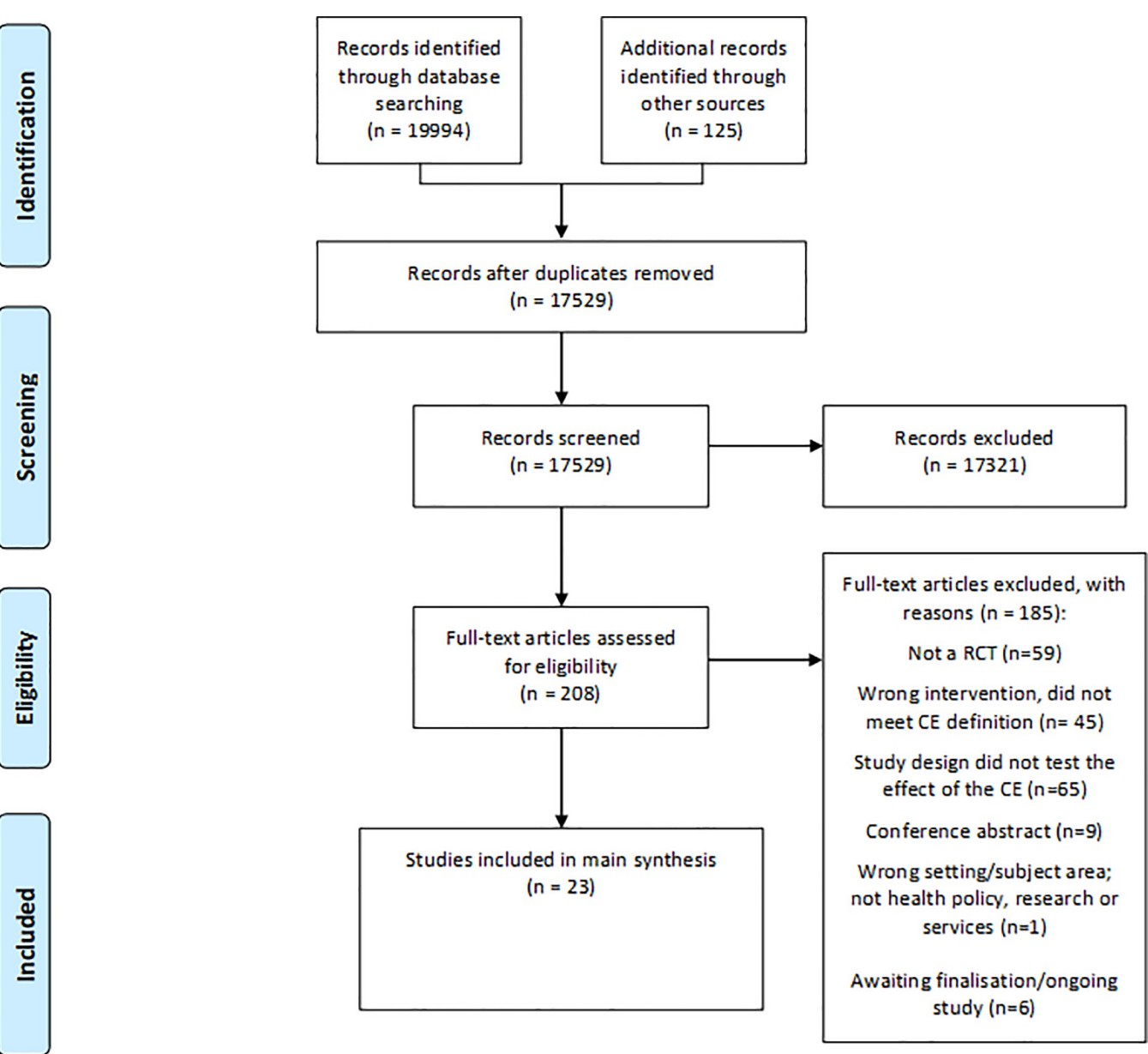

**Fig 1. Modified PRISMA flowchart outlining the search results [41].**

**Table 1. Interventions with consumer engagement as defined by the IAP2 Public Participation Spectrum.**

| | Inform *(not included in review)* | Consult | Involve | Collaborate | Empower |
|---|---|---|---|---|---|
| **Public Participation Goal** | *To provide the public with balanced and objective information to assist them in understanding the problem, alternatives, opportunities and/or solutions* | To obtain public feedback on analysis, alternatives and/or decisions | To work directly with the public throughout the process to ensure that public concerns and aspirations are consistently understood and considered | To partner with the public in each aspect of the decision including the development of alternatives and the identification of the preferred solution | To place final decision making in the hands of the public |
| **Promise to the public** | *We will keep you informed.* | We will keep you informed, listen to and acknowledge concerns and aspirations, and provide feedback on how public input influenced the decision. | We will work with you to ensure that your concerns and aspirations are directly reflected in the alternatives developed and provide feedback on how public input influenced the decision. | We will look to you for advice and innovation in formulating solutions and incorporate your advice and recommendations into the decisions to the maximum extent possible. | We will implement what you decide. |
| **Consumer engagement methods** | · *Mass media*<br>· *Website*<br>· *Press releases*<br>· *Mail outs*<br>· *Fact sheets*<br>· *Hotline*<br>· *Displays and exhibitions*<br>· *Presentations* | · Focus group<br>· Patient surveys<br>· Feedback and complaints<br>· Story-telling<br>· Social media<br>· Planning meetings/forums<br>· Suggestion boxes<br>· Patient diaries<br>· Mystery shopping | · Forums for debate<br>· Health panels<br>· Shadowing patients<br>· Workshops<br>· Public meetings | · Patient advisory councils/committees<br>· Expert patients<br>· Charette (interdisciplinary planning group)<br>· Constituent assembly<br>· Delphi process<br>· Retreats<br>· Round tables<br>· Impact assessments<br>· Ethics committees<br>· World Café<br>· Town hall meetings<br>· Revolving conversation | · Citizen jury<br>· Consumer managed project/service<br>· Consensus conference<br>· Deliberative polling<br>· Search conference<br>· Study circles<br>· Study groups<br>· Sustainable community development<br>· Think tanks |

## Data extraction and management

Two review authors (two of LW, JL, SH) extracted data independently from included studies using a standardised data extraction form. Any discrepancies were resolved by discussion until consensus was reached, or through consultation with other authors where necessary. We piloted the data extraction template to ensure sufficient information about the study design, participants, consumer engagement and study methods, and outcomes measured to inform the interpretation of the results. All extracted data were entered into RevMan (RevMan 2012) by two review authors (JL and SH) and were checked for accuracy against the data extraction sheets by a third review author working independently. In addition to data items to be extracted, the author team assessed the extent of the consumer engagement in each included study. To do this, the method of consumer engagement used in included studies was mapped onto one of the four items of the IAP2 Spectrum of Public Participation (i.e. consult, involve, collaborate or empower), and scored (Yes/No/can't tell) according to the seven principles of the IAP2 quality assurance standard [28] (Table 1).

## Assessment of risk of bias in included studies

We assessed and reported on the methodological risk of bias of included studies in accordance with the Cochrane Handbook [39], which recommends the explicit reporting of the following individual elements for RCTs: random sequence generation; allocation sequence concealment; blinding (participants, personnel); blinding (outcome assessment); completeness of outcome

data; selective outcome reporting; and other sources of bias. Using Covidence systematic review software, two authors (JL, LW) independently assessed the risk of bias of included studies, with any disagreements resolved by discussion with a senior (SH) and consumer author (DK), to reach consensus.

We also contacted study authors for additional information about the included studies when necessary to clarify methods, data or expected completion dates. Studies were categorised as 'awaiting classification' if additional information was not able to be sought, or 'ongoing' if the trial was deemed in progress.

## Data synthesis

We decided whether to meta-analyse data based on whether the interventions in the included trials were similar enough in terms of participants, settings, intervention, comparison and outcome measures to ensure meaningful conclusions from a statistically pooled result.

Where we were unable to pool the data statistically using meta-analysis, we had clear reasons for this decision (outlined in the Results section) and presented the data in tables and narratively synthesised results. We have presented the major outcomes and effects, organised by intervention categories according to the major types and/or aims of the identified interventions. Within the data categories, we explored the main comparisons of the review: consumer engagement to no consumer engagement; one method of consumer engagement to another method of consumer engagement.

If studies compared more than one consumer engagement intervention (e.g. two or more interventions and a no-intervention/control group), we would have compared each intervention separately to the no intervention/control group; and with one another.

## Measures of treatment effect

For dichotomous outcomes, we analysed data based on the number of events and the number of people assessed in the intervention and comparison groups. We used these to calculate the odds ratio (OR) or risk ratio (RR) and 95% confidence interval (CI). For continuous measures, we analysed data based on the mean, standard deviation (SD) and number of people assessed for both the intervention and comparison groups to calculate the mean difference (MD) and 95% CI. If the MD was reported without individual group data, we used this to report the study effects. If more than one study measured the same outcome using different tools, we would have calculated the standardised mean difference (SMD) and 95% CI using the inverse variance method in RevMan 5.

The range of outcomes considered in this review was developed collaboratively with our stakeholder panel and is substantial. Where studies recorded outcome data at multiple time points, we reported the data from the final follow-up, because longer-term outcomes were considered most important in practice by our stakeholder panel. Therefore, we did not report outcome data recorded at other time points.

## Unit of analysis issues

If cluster-RCTs were included, we checked for unit-of-analysis errors. If errors were found, and sufficient information was available, we would have re-analysed the data using the appropriate unit of analysis, by taking account of the intra-cluster correlation (ICC). We planned to obtain estimates of the ICC by contacting authors of included studies or imputing them using estimates from external sources (e.g. similar studies, resources that provide examples of ICCs, or ICC patterns for particular types of cluster or outcome) [39]. If it was not possible to obtain

sufficient information to re-analyse the data we would have reported effect estimates and annotated with a 'unit-of-analysis error' message.

## Assessment of heterogeneity

Where studies were considered similar enough (based on consideration of the populations, consumer engagement activities, or other factors such as outcomes) [42] to allow pooling of data using meta-analysis, we assessed the degree of heterogeneity by visual inspection of forest plots and by examining the $I^2$ test for heterogeneity. We reported our reasons for deciding that studies were similar enough to pool statistically. Heterogeneity was quantified using the $I^2$ statistic. An $I^2$ value of 50% or more was considered to represent substantial levels of heterogeneity, but this value was interpreted in light of the size and direction of effects and the strength of the evidence for heterogeneity, based on the p-value from the $I^2$ test [39]. Where heterogeneity was present in pooled effect estimates we planned to explore possible reasons for variability by conducting subgroup analysis.

## Assessment of reporting biases

We assessed reporting bias qualitatively based on the characteristics of the included studies (e.g. if only small studies that indicate positive findings were identified for inclusion), and if information that we obtained from contacting experts and authors or studies suggested that there were relevant unpublished studies. If we identified sufficient studies (at least 10) for inclusion in the review we planned to construct a funnel plot to investigate small study effects, which may indicate the presence of publication bias. We would formally test for funnel plot asymmetry, with the choice of test made based on advice in Higgins *et al* [39], and bearing in mind that there may be several reasons for funnel plot asymmetry when interpreting the effects.

## Subgroup analysis and investigation of heterogeneity

Given the nature and quantity of trials, it was unlikely we would be able to conduct a formal sub-group analysis; we therefore explored any potential effect modifiers narratively. We considered whether the differences in any of the following factors might explain any differences in the effects:

- the methods of consumer engagement per IAP2 participation level [28] (recorded verbatim from included studies; e.g. focus groups and patient surveys, patient advisory committees and Delphi processes, and citizen panels or consumer managed projects and services) [23, 35];

- IAP2 Public Participation Spectrum (with the four items: consult, involve, collaborate, and empower) [28], the stages (i.e. development, implementation, monitoring, evaluation) of health care policy, research and services in which consumers are engaged;

- differences between consumer or professional participants; and

- context (health care policy, research, services).

## Sensitivity analysis

Where meta-analysis was possible, we considered a sensitivity analysis to determine the effects of including studies at highest risk of bias. This meant comparing the effects of studies deemed at highest risk of bias (as outlined in *Assessment of risk of bias of included studies*) with those rated at lower risk of bias. By default, the highest risk of bias studies would include quasi-RCTs.

## Results

### Description of included studies

The prior reviews [23] reported on a total of six included trials. The revised and updated searches for the 2020 version identified an additional 14,540 citations, with full-text copies of 156 articles retrieved for further assessment. Seventeen new trials, described in 41 papers, were then added to the six trials from the original review (Fig 1 for flow chart of search results). We identified 16 trials through database searches, one from hand searching the reference lists of relevant publications, and three ongoing studies from searches of trial registries (ISRCTN41083256; NCT02319967; KT Canada 87776; ACTRN12614000457640).

Twelve inclusions were RCTs, and the remaining 11 were cluster RCTs [43–53]. Six studies included 100 or fewer participants [11, 45, 52, 54–56], nine studies had between 100 and 1,000 participants, and eight had more than 1,000 participants including three studies with over 20,000 participants [44, 47, 51]. Study authors (n = 4) were contacted via email (up to a maximum of three occasions over a two-month period) to clarify: (i) study completion dates (for published study protocols and pilot/foundational studies that indicated the main study was still in progress), and (ii) details of the consumer engagement strategies (especially how they were developed) to determine if they met our eligibility criteria. Author contact for the latter was not especially helpful as methodological ambiguity most often arose from older studies, the authors of which typically did not reply.

### Participants and setting

Included studies were conducted in ten countries (USA, UK, Canada, Bangladesh, Norway, Ghana, Japan, Nepal, Vietnam, Belgium), in both urban and rural settings. Nine studies were set in (self-described) low socioeconomic countries and/or within disadvantaged communities [43, 44, 46, 47, 50, 51, 53, 56, 57]. The interventions for most studies took place within outpatient [57–60 or local community health clinics [43, 45, 49], or social support hubs [44, 47, 50, 51], and two studies were conducted in hospital inpatient settings [55, 61], one in a medical school [62], and one in aged care facilities [52].

The interventions were directed at people within various diagnostic groupings including mental illness [49, 53, 56, 59, 60]; pregnant and birthing women [44, 47, 50, 51]; and chronic diseases [45, 58] including smoking [46]. Other targeted participants were users of health care services such as participants undertaking hospital procedures [55, 61], attending Oncology clinics [63], or attending Paediatric clinics [57]. Other participants represented general health care users in their community [43, 54, 64], military veterans [48], nursing home residents [52], advocates for patients with Alzheimer's and their caregivers [11], and medical students [62].

### Interventions

To describe the interventions involving consumer engagement, we classified trials according to the area (context), level (methods), stage and characteristics of the consumers. Table 2 provides a summary of these features.

**Areas of consumer engagement.** The majority of interventions were in the area of health **services** [15], predominantly in the development (and/or implementation) of the service itself; although two were more specifically involved in developing patient information material [55, 61], two in developing training material for health professionals who deliver health services [62, 63]. Four studies engaged consumers in health-related **research**, spanning development, implementation and monitoring within the research process [48, 49, 59, 60]. The final four

**Table 2. Results–included trials, with summary of population, intervention details, comparisons, outcomes and findings.**

| Study and population | Area of CE | Level of CE | Stage of CE | Outcomes | Findings |
|---|---|---|---|---|---|
| Aabakken 1997 [61] Endoscopy patients (n = 235) | Health Services | Consult | Development | Levels of anxiety Patient satisfaction | Favours CE No difference |
| Abelson 2003 [54] Community groups CE1vs CE2 vs CE3 (n = 46) | Health Policy | Collaborate | Development | Prioritising health concerns Rating importance of strengths Ranking health determinants | CE priorities more likely to change CE greater environment/local education CE greater employment/conditions |
| Alhassan 2016 [43] Health staff (n = 234 staff; 64 health facilities) | Health Services | Collaborate | Monitoring Evaluation | Safe-care essentials–patient safety, quality etc Motivation levels | Favours CE (overall risk score p<0.05) No difference for some sub items Favours CE (overall motivation score p<0.0001) |
| Armstrong 2018 [11] Guideline development groups (n = 18 participants) | Health Policy | Collaborate | Implementation | Descriptive comparison of proposed PICOT questions, benefits, and harms between groups Qualitative analysis of discussion themes from audio recordings of the question development retreat discussions | Proposed guideline questions, benefits and harms largely similar between groups Only the CE group proposed outcomes related to the importance of being able to plan for the future. CE influenced the conduct of guideline development, scope, inclusion of patient-relevant topics, outcome selection, and planned approaches to recommendation development, implementation, and dissemination. |
| Azad 2010 [44] Bangladesh women giving birth (n = 43 717 births) | Health Services | Empower | Development Implementation | Neonatal mortality rate Maternal death Health services | No difference Favours no CE (RR1.91; 95%CI 1.27,2.9) No difference |
| Boivin 2014 [45] Patients with chronic conditions (n = 17 patients, 44 professionals, 6 cluster sites) | Health Policy | Involve | Development | Level of agreement between patient and professional priorities Changes in priorities/prof intentions/cost | CE priorities in agreement Different changes–CE more community; prof more technical No difference in intentions or cost |
| Carman 2015 [64] Health consumers CE1 vs CE2 vs CE3 vs control (n = 1774, 76 groups) | Health Policy | Collaborate | Development | Participant knowledge Attitudes toward decision-making Attitudes towards hospital use | Favours any CE (p<0.05) vs no CE Varied attitudes mostly not different |
| Choi 2016 [46] American Indians who smoke (n = 624) | Health Services | Collaborate | Development | Smoking abstinence | Favours CE at 12 weeks and 6 months for self-report of quitting No difference for salivary tests |
| Chumbley 2002 [55] Surgical patients (n = 100) | Health Services (PIM) | Consult | Development | Clarity and knowledge of PCA Worries about using PCA | Favours CE for clarity of information and knowledge of PCA; no difference for worries |
| Clark 1999 [59] Patients with mental health diagnosis (n = 120) | Health Research | Collaborate | Implementation Monitoring | Patient satisfaction Negative responses Positive responses | No difference Favours CE No difference |
| Coker 2016 [45] Children (lower income) with parent coach (n = 251) | Health Services | Consult | Development | Receipt of services Parent experiences Service use–healthcare utilisation | Favours CE Favours CE Aspects in favour; others no different |
| Corrigan 2017 [56] Homeless with mental health diagnosis (n = 67) | Health Services | Empower | Implementation | TCU-HF–health status and QoL Homelessness SF-36; Recovery scale; QoL Scale | All outcomes favour CE |
| Early 2015 [58] Respiratory outpatients (n = 165) | Health Services | Consult | Development | Satisfaction; Confidence; Outcome; Consultation time | All outcomes no difference |
| Fottrell 2013 [47] Bangladeshi women giving birth (n = 13459 pregnancies/ neonatal events) | Health Services | Empower | Development Implementation | Neonatal mortality Maternal death rate; process indicators; maternal psychology | Favours CE (OR 1.91; 95% CI 0.55,0.8) No difference or qualified differences only for other outcomes |
| Fujimori 2014 [63] Oncologists having communication training (n = 601 patients, 30 oncologists) | Health Services (training) | Consult | Development | Objective performance Confidence communication Patient distress (HADS A and D); satisfaction; trust | Favours CE Favours CE Favours CE for patient trust/depression; No difference for anxiety or satisfaction |

*(Continued)*

**Table 2.** (Continued)

| Study and population | Area of CE | Level of CE | Stage of CE | Outcomes | Findings |
|---|---|---|---|---|---|
| Guarino 2006 [48]<br>Military veterans (n = 1092, 10 medical centres) | Health Research | Involve | Development | Participant understanding<br>Satisfaction<br>Adherence/participation | No difference for all outcomes |
| Hughes-Morley 2016 [49]<br>Recruitment into mental health trial (n = 8182 patients) | Health Research | Involve | Development | % recruited through CE<br>% responded/recruited with telephone follow-up | No difference<br>No difference |
| Jha 2015 [62]<br>Junior Drs in patient safety (n = 283) | Health Services (training) | Collaborate | Development Implementation | Attitude to patient safety<br>+ve & -ve affect scales | No difference for attitudes to safety<br>Favours CE for +ve affect |
| Manandhar 2004 [50]<br>Nepalese women giving birth (n = 28931, 6380 pregnancies) | Health Services | Empower | Development Implementation | Neonatal mortality rates<br>Maternal deaths<br>Still births; Service uptake; Home care practices; Infant morbidity | Favours CE (OR 0.7; 95%CI 0.52, 0.94)<br>Favours CE (OR 0.2; 95%CI 0.04, 0.91)<br>No difference except for uptake of services favours CE |
| Persson 2013 [51]<br>Vietnamese women giving birth (n = 22561 births) | Health Services | Empower | Development Implementation | Neonatal mortality rates<br>Frequencies live/still births; staff knowledge; resources and usage | No difference overall (favours CE 3rd year)<br>No difference except for increased care usage favours CE |
| Polowczyk 1993 [60]<br>Patients with mental health Dx (n = 530) | Health Research | Involve | Implementation Monitoring | Patient satisfaction | Favours control (no CE in treatment) |
| Van Malderen 2017 [52]<br>Residents of ACF<br>CE1 vs active vs passive controls (n = 88) | Health Services | Collaborate | Development Implementation Monitoring Evaluation | Active ageing survey;<br>QoL;<br>Participation and autonomy scale | No difference<br>Favours CE and active control<br>No difference |
| Wells 2013 [53]<br>Patients with mental health diagnosis (n = 1246) | Health Policy and Services | Collaborate | Development Implementation | Mental health scale<br>Services use; socioeconomic factors | Favours CE all items<br>Favours CE except no difference for employment and use of medication. |

Key: CE = consumer engagement; CI = confidence interval; PICOT = Population, Intervention, Comparator, Outcome, Time; QoL = quality of life; OR = odds ratio; PCA = patient controlled analgesia; RR = relative risk; SF-36 = Short Form (36) Health Survey; TCU-HF = Texas Christian University Health Form.

trials employed consumer engagement in the area of health **policy**, predominantly in developing priorities [45, 53, 54, 64] but also in the development of a clinical practice guideline [11].

**Levels of consumer engagement.** Using the IAP2 descriptors of consumer engagement, five trials were judged to be **consulting** [55, 57, 58, 61, 63], four were **involving** [45, 48, 49, 60], nine were **collaborating** [11, 43, 46, 52–54, 59, 62, 64] and five were **empowering** [44, 47, 50, 51, 56].

**Stages of consumer engagement.** Consumer engagement occurred mostly at the stage of **development** (n = 18), with 11 at the stage of **implementation**, four **monitoring** and one **evaluating**. Ten trials engaged with consumers at two or more stages (Table 2).

**Characteristics of consumers.** In all trials, the consumer engagement participants had the relevant background as the intended recipient participants—whether that was related to a particular cultural or ethnic characteristic, or diagnostic or sociodemographic grouping (e.g. age or community-dwelling or service user). No consumers in the engagement process were reported to have had prior training in engagement, but the majority received relevant training as part of the trial process (13 trials) [43–47, 50–53, 56, 57, 59, 60] which ranged from an hour or two familiarising the consumers with the task at hand, through to several days spent in training, discussion and development.

## Outcomes

Table 2 summarises the outcome measures used across the trials. Twenty-one trials [43–53, 55–64] considered the effects on the intended recipient participants; only one trial [48]

measured the effects of the intervention on the engagement participants themselves; five [11, 45, 52, 54, 57] measured the effects on research, policy or health care service itself and nine [11, 43–47, 50, 51, 53] measured process outcomes.

## Excluded studies

Of the 180 full texts excluded in this update, 43 studies used an intervention that did not fit our definitions for consumer engagement, and 58 used the wrong study design (i.e. were not RCTs), nine were conference abstracts, and one did not relate to health research, policy or services (i.e. wrong subject area). The most common reason for exclusion (n = 64 studies) was the use of a design that could not differentiate the effect of consumer engagement intervention from other effects (for a full list please contact authors).

## Risk of bias in included studies

Overall the risk of bias in the included studies was high, confounded by poor (unclear) reporting particularly in the earlier studies as noted in the 2006 and 2009 reviews; on average, there were 6.3 items with unclear risk of bias ratings for the 7 studies published between 1997 and 2007), compared with 2.3 items for those (n = 16) from 2008 onwards. Only two studies achieved four criteria as low risk [45, 49]; the average number of criteria judged as low risk was only 1.3 per trial. Fig 2 shows individual ratings for each risk category.

## Effects of interventions

Twenty-one [11, 43–53, 55–63] of the 23 included trials were comparisons of consumer engagement versus no consumer engagement. Two trials involved comparisons of different methods or degrees of consumer engagement, all compared with no consumer engagement [54, 64]. No trials were identified that only compared one type of engagement with another. Subgroup analyses were not able to be performed with insufficient numbers of trials in any sub-group of interest, as expected.

## Consumer engagement versus no consumer engagement in health services (15 included trials)

**Health service delivery.** We were able to find four studies [44, 47, 50, 51] sufficiently similar across the PICO domains to allow meta-analyses. These studies all investigated women giving birth who received a comprehensive health services intervention for healthy birthing that was developed and implemented with consumer engagement at an *empowering* level, compared to birthing education and support developed and delivered without consumer engagement (across three different countries) [44, 47, 50, 51]. All four trials used cluster randomisation and all four performed appropriate analyses at both the individual and cluster levels. All stated their intent to account for the cluster design in their protocols and all carried this intention out in the final reports; therefore, no unit of analysis issues were present in the outcome data. For the outcomes of neonatal mortality, we extracted data for 83,925 births, with a total of 1,028 deaths for the intervention group and 1,282 for the control group; this gave a meta odds ratio of 0.8 in favour of the intervention group (95% CI 0.77, 0.91; p<0.0001) (Fig 3) (or Risk Ratio of 0.84; 95% CI 0.77, 0.91; p<0.0001). The heterogeneity (as assessed by the $I^2$) was high at 72% and mostly explained by the large range of frequencies of event. The second outcome (maternal deaths) that afforded meta-analysis returned a non-significant finding from the same four studies [44, 47, 50, 51] and same number of births with 80 maternal deaths in the intervention group versus 78 in the control (OR of 1.10, 95% CI 0.81,1.51;

| | Random sequence generation (selection bias) | Allocation concealment (selection bias) | Blinding (performance bias and detection bias) | Blinding of participants (performance bias) | Blinding of outcome assessment (detection bias) | Incomplete outcome data (attrition bias) | Selective reporting (reporting bias) | Other bias |
|---|---|---|---|---|---|---|---|---|
| Aabakken et al. [61] 1997 | ? | ? | ? | ? | ? | ? | ? | ? |
| Abelson et al. [54] 2003 | ? | ? | ? | ? | ? | ? | ? | − |
| Alhassan et al. [43] 2016 | − | − | − | − | − | + | ? | − |
| Armstrong et al. [11] 2018 | ? | + | − | − | − | + | + | + |
| Azad et al. [44] 2010 | + | − | − | − | − | + | ? | − |
| Boivin et al. [45] 2014 | + | + | − | − | ? | + | + | − |
| Carman et al. [64] 2015 | − | ? | − | − | ? | ? | − | − |
| Choi et al. [46] 2015 | + | + | − | − | − | ? | − | + |
| Chumbley et al. [55] 2002 | + | ? | ? | ? | ? | − | − | ? |
| Clark et al. [59] 1999 | ? | ? | ? | ? | ? | ? | ? | ? |
| Coker et al. [57] 2016 | + | − | − | − | − | + | ? | ? |
| Corrigan et al. [56] 2017 | ? | ? | − | − | ? | − | ? | − |
| Early et al. [58] 2015 | + | − | − | − | ? | ? | ? | ? |
| Fottrell et al. [47] 2013 | − | − | − | − | − | ? | ? | − |
| Fujimori et al. [63] 2014 | ? | ? | − | − | − | ? | ? | − |
| Guarino et al. [48] 2006 | ? | ? | ? | ? | ? | − | ? | − |
| Hughes-Morley et al. [49] 2016 | + | + | − | ? | ? | + | ? | − |
| Jha et al. [62] 2015 | + | ? | − | − | ? | − | − | ? |
| Manandhar et al. [50] 2004 | ? | − | ? | − | ? | + | ? | − |
| Persson et al. [51] 2013 | + | + | − | − | − | ? | + | − |
| Polowzyk et al [60] 1993 | ? | ? | − | − | ? | ? | ? | ? |
| van Malderen et al. [52] 2017 | ? | ? | − | ? | + | ? | ? | + |
| Wells et al. [53] 2013 | ? | ? | − | − | + | + | ? | − |

**Fig 2. Risk of bias of included studies.**

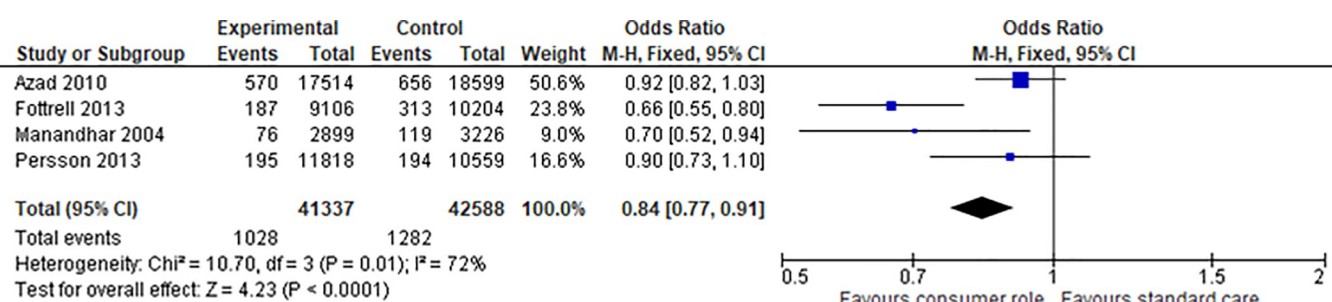

**Fig 3. Meta-analysis of consumer engagement interventions for the outcome of neonatal mortality.**

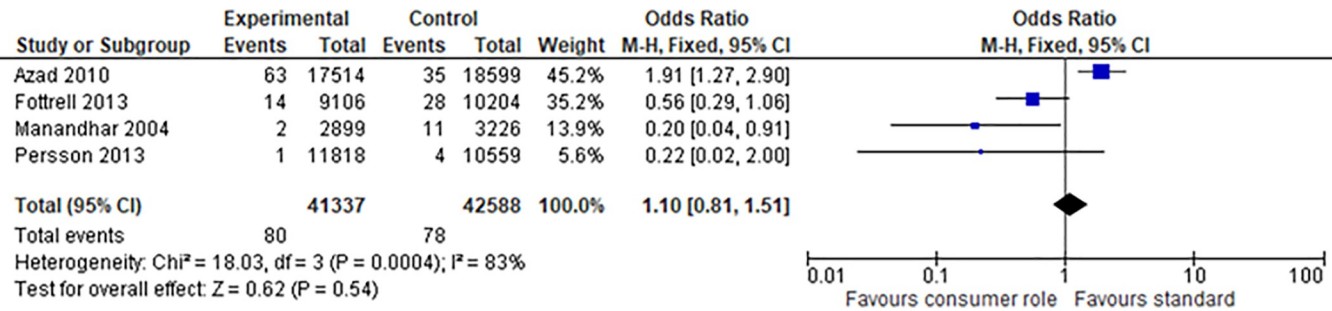

**Fig 4. Meta-analysis of consumer engagement interventions for the outcome of maternal deaths.**

p = 0.54) (Fig 4). A risk ratio analysis revealed the same results. Again the heterogeneity was high at 83% and this time was predominantly explained by the result of one study [44] which reported an unforeseen maternal death rate of 63 in the consumer engaged group versus 35 in the control; in addition, heterogeneity by chance may have been exaggerated by the small number of events (maternal deaths). The authors could not explain the reason for this and felt it was not attributable to the intervention as there was no common cause of death. The other three studies [47, 50, 51], in contrast, all reported lower maternal death rates in the intervention groups (two of the three non-significant).

Based on these four trials [44, 47, 50, 51] there is evidence that consumer engagement in **developing and implementing health services**, to enhance the care of mothers and their babies, results in a reduction in neonatal, but not maternal, mortality.

Three other trials [46, 57, 58] investigated the effect of consumers in **developing health services**. Choi *et al* [46] collaborated with consumers from a culturally appropriate community group to develop community-based smoking cessation program, finding in favour of increased self-reported quitting for the consumer engagement intervention, but no effect on saliva-based testing for smoking abstinence. Coker *et al* [57] consulted parents to develop parent coaches for children from lower-income groups, and reported favourable effects on improved receipt of services and satisfaction with consumer engagement-developed services compared to non-consumer engagement. Early *et al* [58] consulted with consumers in the development of respiratory outpatient services but found no difference in outcomes between the services that were consumer-consulted versus not. Based on these findings there is some evidence that some aspects of health service development may be improved by consumer engagement, but the evidence is not clear as to what aspects these may be in any consistent way.

One trial [56] investigated the effect of consumer engagement on implementation only of health services. Corrigan *et al* [56] used an empowerment model with consumers to implement services for people who were both homeless and with mental health issues. They reported that all outcomes were in favour of the consumer engagement-empowered services including health status, quality of life, and rates of homelessness. One trial [43] collaborated with consumers in the monitoring and evaluation of health services for safety, quality and health staff motivation. They reported that patient safety and staff motivation improved with the consumer-collaboration processes, and no change for quality items. Van Malderen *et al* [52] collaborated with residents of an aged care facility to develop, implement, monitor and evaluate several training interventions versus an intervention with no consumer input. They reported no difference between the various interventions except that the consumer-collaboration interventions favoured improved quality of life.

**Patient information.** Two trials evaluated products (patient information leaflets about endoscopic procedures [55] and post-operative patient-controlled analgesia (PCA) [61])

which were developed following consumer consultation. The leaflets were compared with patient information developed without consumer consultation. Based on these two trials [55, 61] evidence is equivocal for consumer consultation prior to developing patient information material; while the CE material resulted in better anxiety outcomes for endoscopy patients in one study (no effect for satisfaction) [55], the other showed that despite the CE material presenting information more clearly and improving patient knowledge of PCA, there was no effect for worry about PCA-use [61]. Both studies had unclear risk for all biases (i.e. not stated within methodology) except for low risk of selection bias (use of random allocation) [61], and high risk for attrition (not all outcome data reported, no reasons given) and selective reporting bias [61] (Fig 2).

**Health professional training.**   Two trials [62, 63] engaged consumers in the development of education material for medical practitioners. Fujimori *et al* [63] consulted with consumers in the development of communication training with oncologists and reported that consumer-based training led to improved objective performance and confidence in communication by the doctors and possible improvement in patient trust, though there were no differences for patient anxiety or satisfaction. Jha *et al* [62] collaborated with consumers to develop and implement training for junior doctors in patient safety and reported no difference in junior doctors' attitudes to safety but favourable changes in their positive affect. Based on these two trials [62, 63] there are mixed effects of consumer-engagement (consultation) on the development and/or implementation of health professional training.

## Consumer engagement versus no consumer engagement in health care policy (five included trials)

Two trials [45, 54] engaged consumers in priority setting processes for health services. Abelson *et al* [54] compared three different methods of consulting consumers, reporting that consumers did have some different perspectives with a greater focus on environmental, local education and employment conditions. Boivin *et al* [45] involved consumers in setting priorities for chronic disease prevention and management, finding that consumers and professionals were in broad agreement although differed on specifics; for example, consumers were more focused on community whilst the professionals were more focused on the technical aspects. Carman *et al* [64] collaborated with consumers in exploring ethical decision-making in health care—they compared different ways of deliberating with the consumer group via face to face or online options compared to no consumer engagement and reading material only, and found that all options which involved consumer engagement were superior to the control group for knowledge and attitudes towards decision-making, as well as attitudes to hospital use. Based on these studies there is some evidence that consumers may have a role in identifying a broader range of health care priorities that are complementary to those identified by professionals [45, 54]. In addition to supporting decision-making processes in health (reflected in public deliberation increasing participants' knowledge of and attitudes towards the role of medical evidence) [64], there is also some evidence of consumers contributing positively to identifying need and developing mental health service directions [53]. The fourth policy-based trial [53] collaborated with consumers with previous mental health issues in identification, planning, development (and implementation) of community-based services. The services developed with consumer collaboration were superior to services developed with no consumer engagement in all measured items (mental health scores, service use and so forth), except employment and medication use.

The remaining policy-related trial [11] compared two guideline development groups, one collaborating with consumers and one without, to craft Population, Intervention, Comparator,

Outcome, Time (PICOT) questions, and identify key benefits and harms on the topic of using amyloid positron emission tomography in the diagnosis of dementia. "The proposed guideline questions, and benefits and harms, were largely similar between the two groups" [11 p.1]; however, only the group that collaborated with consumers "proposed outcomes around the future development of cognitive impairment at certain time points and proposed rate of progression (rather than considering the development of dementia as a binary [yes/no] outcome) [11 p.9-10]. Armstrong et al [11] reported that consumer collaboration also influenced the "conduct of guideline development, scope, inclusion of patient-relevant topics, outcome selection, and planned approaches to recommendation development, implementation, and dissemination" [11 p.11].

## Consumer engagement versus no consumer engagement in research (four included trials)

Overall four studies investigated consumer engagement in the conduct of research–three of which were in the earlier review by Nilsen *et al* [23]. As reported in the first review, two trials compared consumers (former patients) with professionals as data collectors in patient satisfaction surveys in mental health services [59, 60], Both studies found that participants reported high levels of satisfaction with mental health outpatient services irrespective of interviewer. However Clark *et al* [59] found that consumer interviewers elicited significantly more 'extreme negative' responses, compared to responses gained by staff interviewers (p = 0.02). Polowczyk *et al* [60] also found that the consumer (client) interviewed group on average gave lower satisfaction scores than in the staff interviewed group (0.16 on a scale from 1 to 4, p = 0.05) [60]. Nilsen *et al* pooled the results of these two trials [59, 60] finding the overall difference was similar (0.14 on a scale from 0 to 4, p = 0.001), (Fig 5) [23]. Their summary remains appropriate: based on these same two trials there are small differences in satisfaction survey results when consumer interviewers are used instead of staff interviewers.

Two trials [48, 49] engaged with heath researchers in other research processes. Guarino *et al* [48] (included in the earlier review [23]) compared an informed consent document developed with consumer involvement (potential trial participants) to a consent document developed by professionals (trial investigators). They found no overall difference in understanding between the two groups. Hughes-Morley *et al* [49] (new in this review) involved consumers in recruiting participants for a mental health care trial and found no difference between the number recruited using consumer engagement-based processes versus non-consumer engagement involved methods. Despite the addition of this second trial [49], there is no change to the results reported in Nilsen et al [23]: consumer consultation in the development of consent documents or recruitment may have little, if any, impact on trial participant's self-reported understanding, satisfaction, adherence or recruitment.

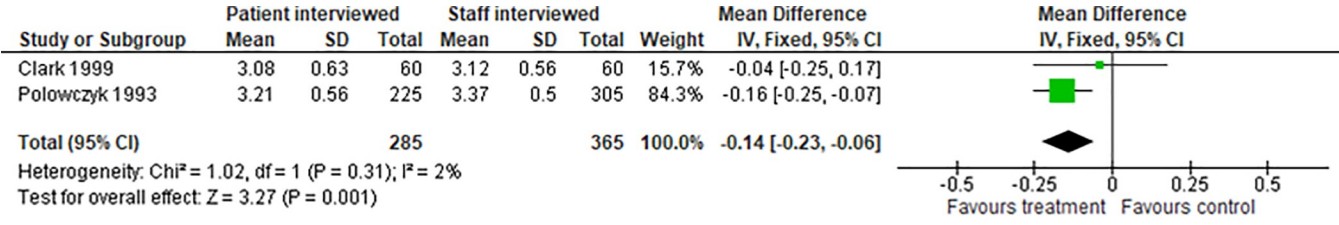

**Fig 5. Meta-analysis of consumer engagement interventions for the outcome of satisfaction.**

## Discussion

### Statement of principal findings

The primary objective of this updated review was to assess the effects of consumer engagement on health care policy, research and services. There has been a rapid increase in the number (and, in some instances, quality) of randomised controlled trials investigating the effectiveness of consumer engagement. The most noteworthy rise in the number of trials is in the area of health services (up from 2 to 15), with only a single addition in each of the policy and research domains. Typically, the trials compared a form of engagement with no engagement. The most common objective of the health service trials was to improve patient outcomes and the primary meta-analysis gave evidence that empowering consumers in the development and implementation of community-based services for pregnant women can reduce neonatal mortality, although there was no effect on maternal death rates. Coupled with other single trials, there is emerging evidence that consumer engagement in health service delivery affords benefits (and no evidence of harms to date) and these seem to be particularly evident in community health care settings. It should also be noted that the outcomes of the health services trials were wide ranging beyond health effects on the intended trial participants and included satisfaction, health behaviour adoption, knowledge and service utilisation. There is some evidence that consumers can have a positive role in the training of health professionals and in providing information for patients to inform their decision-making, although there is still no trial evidence for the role of consumers in guideline production.

Investigations into the effect of consumer engagement in health policy were less prevalent but did include some trials demonstrating potential benefits of collaborating with consumers in health priority-setting where they can broaden the perspectives otherwise gained from health professionals alone. There remains a dearth of investigation into the effect of consumer engagement in policy decision-making. The investigations of the effect of consumer engagement in health research remains limited to process indicators like recruitment, adherence, information and satisfaction, with no trials investigating the benefits of consumers being engaged in setting a research agenda or research methodology.

Secondary objectives were to explore whether differences between studies might explain any differences between the effects. We were specifically interested in differences in the:

- methods (levels) of consumer engagement (e.g. fact sheets, focus groups, patient advisory committees);

- stages (i.e. development, implementation, monitoring, evaluation) of health care policy, research and services in which consumers are engaged; and

- characteristics of consumer or professional participants (e.g. background, experience or training in consumer engagement).

Such a broad range of trials did not allow for rigorous investigation of the influence of these differences using sub-group analyses. However, there are patterns that warrant commentary. Firstly the strongest evidence (community-based antenatal care for pregnant women) lay in the **method of engagement** that lies within the definition of *empowerment*—using the definitions from the IAP2 of placing "the final decision-making in the hands of the public" and with the tacit understanding "we will implement what you decide" [28]. The four studies [44, 47, 50, 51] that provided the evidence for improved neonatal mortality all *empowered* consumers in this decision-making, with the consumer supported project and sustainable community development. Other key characteristics were that the local community (i.e. women's groups, community leaders and member, locally recruited facilitators) were embedded in every phase of

the CE program; in developing its content, its implementation plan, and its delivery. The majority of the remaining studies used *collaborative* methods such as advisory groups, special meetings and committees; and smaller numbers using *involving* methods (via public meetings) or *consulting* (via focus groups and surveys), Tables 1 and 2.

The predominant **stage of consumer engagement** was in *development* (over 75%), with a third of these [44, 47, 50, 51, 53, 62] (including the four comprehensive health service interventions for healthy birthing) [44, 47, 50, 51] undertaking engagement in both *development AND implementation*. This seems a logical paired process and again was most common in the community settings, to ensure buy-in and relevance for service delivery in most examples. Only one trial [52] collaborated with consumers at all stages from development through to evaluation. However, the numbers are too small to allow a statistical analysis of the import of this factor on effectiveness.

Finally, we considered the **characteristics of the consumers** in the trials. Again, there was a consistent finding that the consumers were not formally trained in engagement methods prior to the engagement, but the majority received requisite training in the study methodology or intervention to be delivered as part of the engagement process. In three studies [52, 59, 60], health professional participants were reported to receive this same training. Only eight trial consumer cohorts [11, 49, 54, 55, 58, 61, 62, 64] received no training before or during the trial —bringing their lived experience as sufficient expertise for the purposes of the evaluation. This is an important finding for future trials—to consider the engagement of all participants–professionals and consumers–in health care stages across policy, research and services.

We found no evidence of adverse effects in engaging with consumers; most trials did not collect data on this aspect. The trials with patient health level data (such as neonatal mortality) [44, 50] considered unintended effects in terms of examining the risk ratios for all outcomes between the two groups. None of the included trials addressed possible other adverse effects of consumer engagement, such as tokenism or time and other resource impacts. Seven studies [43, 45–47, 50, 51, 53] did consider cost-effectiveness, reporting favourably on the cost per life saved and/or cost of years of life saved in the case of the neonatal care studies, or the cost of the intervention itself, but with no benefits or effectiveness analysis [53].

## Strengths and weaknesses of the study

By including a strong stakeholder engagement at key stages in the review process we believe we have mitigated against any major biases.

Our included trials were wide ranging internationally and across diverse health populations. In particular, we were struck by the number of trials engaging with consumers who were most likely to experience disparity or disadvantage in health services. The settings were predominantly in the community—this seems to lend itself to participatory processes. However, there were some examples of consumer engagement in acute and outpatient settings that augur well for ongoing investigations across the health services spectrum. Positive findings across this range suggests that consumer engagement has a role in many settings, however this does not negate the need for consideration to always be paid to these contextual factors when developing, implementing, monitoring and evaluating consumer engagement processes.

Whilst we note the increase in number of more strongly designed (randomised controlled) trials in this important area, most still had a relatively high risk of bias. It should be acknowledged that it can be hard to blind participants–consumers and professionals–in a health care intervention. However, more achievable aspects like adequate randomisation and allocation should be more stringently adhered to, as well as better reporting of outcomes and attrition.

## Strengths and weaknesses in relation to other studies

Treweek *et al* [65] published a Cochrane review regarding strategies to improve recruitment to randomised trials. Their key results relating to consumer engagement in recruitment are similar to the results we have found with respect to patient information; in that consumer involvement in developing the content, format and appearance of information leaflets for potential research participants resulted in only a 1% absolute improvement in recruitment (95% CI: −1% to 3%) [65 p.2].

Crawford *et al* [6] published a systematic review with the aim of examining the effects of involving patients in the planning and development of health care. They reported a low level of studies (mostly case studies) which support the notion that involving patients does contribute to changes across a range of settings, however they found no evidence base for the effects of this involvement on various outcomes such as health, service usage and/or quality.

## Meaning of the study: Possible explanations and implications for clinicians and policymakers

Engaging consumers can have a positive effect in health care policy, research and services, however there are no standard metrics to guide and benchmark evaluation of this effect across settings and contexts. The factors that underpin successful consumer engagement can be hypothesised from the included trials as involving consumer **empowerment** processes, in the **development** and **implementation** phases; there is a lack of evidence arising from the trials regarding the effect of other elements of engagement (inform, consult, involve, collaborate and empower) across all phases. Our recommendation is to use a tailored approach to consumer engagement with an IAP2 level that is as high and appropriate to the goals and promises to the public, and embedded evaluation.

The effect of engaging consumers in health care policy, research and services is gaining attention in the literature. The 23 trials included in this review demonstrate that randomised controlled trials of consumer engagement are feasible. Variation in practice reflects the complex nature of consumer engagement and a climate of innovation rather than evidence-based practice. The evidence from this review suggests that the best methods to achieve effective engagement are likely to vary and will need to engage participants (professionals and consumers). It is our contention that more work needs to be done to implement consumer engagement strategies and solutions specific to each problem. For instance, metrics used in assessing effectiveness and overall quality improvements associated with a health care service delivery project may be very different to that of a clinical trial, versus that of a more translational research-based project. The small number of studies available have forced the comparison of methods and outcomes across different contexts and metrics. Trials are needed to evaluate the effects of different methods of:

- identifying participants (health consumers and professionals)

- determining and utilising the elements and phases of engagement

- participant engagement training and support

- distinguishing purposefully between consumers with lived experience versus the community at large (acknowledging that health professionals have a unique and privileged knowledge of the health sector and therefore, while consumers of health policy, research and services, cannot be seen to contribute a consumer voice representative of the wider community)

- timing the engagement (which included studies did not evaluate)

- engaging individuals, groups and organisations

- resourcing engagement including customised models of financial support

- evaluation.

## Conclusion

There is growing evidence from randomised controlled trials of the effects of consumer engagement on the relevance and positive outcomes of health policy, research and services; however, there are no standard metrics to guide evaluation of this effect. The factors that underpin successful consumer engagement can be hypothesised from the included trials as involving consumer empowerment processes, in the development and implementation phases. Our recommendation is for health care consumers, providers, researchers and funders to continue to employ evidence-informed consumer engagement in their jurisdictions, using a tailored approach with an IAP2 level that is as high and appropriate to the goals and promises to the public, and embedded evaluation.

## Supporting information

**S1 Appendix. Stakeholder panel members and engagement process.**
(DOCX)

**S2 Appendix. Key definitions adopted in the review.**
(DOCX)

**S3 Appendix. Medline search strategy.**
(DOCX)

**S4 Appendix. Reporting guideline checklist–PRISMA.**
(DOCX)

## Acknowledgments

The authors would like to thank Ms Anneliese Synnot and Ms Anne Parkhill for their assistance in this systematic review.

The University of South Australia provided salaries for the author SH, and via internal grant funding for LW, JL and AW. We are very grateful for the volunteer work of DK and the stakeholder group.

## Author Contributions

**Conceptualization:** Louise K. Wiles, Debra Kay, Julie A. Luker, Ellen Kerrins, Susan L. Hillier.

**Data curation:** Louise K. Wiles, Julie A. Luker, Susan L. Hillier.

**Formal analysis:** Louise K. Wiles, Debra Kay, Julie A. Luker, Susan L. Hillier.

**Investigation:** Louise K. Wiles, Debra Kay, Julie A. Luker, Anthea Worley, Susan L. Hillier.

**Methodology:** Louise K. Wiles, Debra Kay, Julie A. Luker, Anthea Worley, Susan L. Hillier.

**Writing – original draft:** Louise K. Wiles, Debra Kay, Julie A. Luker, Susan L. Hillier.

**Writing – review & editing:** Louise K. Wiles, Debra Kay, Julie A. Luker, Anthea Worley, Jane Austin, Allan Ball, Alan Bevan, Michael Cousins, Sarah Dalton, Ellie Hodges, Lidia Horvat,

Ellen Kerrins, Julie Marker, Michele McKinnon, Penelope McMillan, Maria Alejandra Pinero de Plaza, Judy Smith, David Yeung, Susan L. Hillier.

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
