## [Decision Letter · Decision Letter 0]

17 Mar 2021

PONE-D-20-36087

Consumer engagement in health care policy, research and services: methods and effects

PLOS ONE

Dear Dr. Wiles,

Thank you for submitting your manuscript to PLOS ONE. After careful consideration, we feel that it has merit but does not fully meet PLOS ONE’s publication criteria as it currently stands. Therefore, we invite you to submit a revised version of the manuscript that addresses the points raised during the review process.

The reviewer has requested some revisions, in addition to the items raised by the reviewer, please address the following points before more consideration:

Consumer (Or an illustrative term Customer) is a crucial part of any system success, and play a determinant role in policy/ program success and quality. There are a rigorous relationship between customer participation in service delivery and quality improvement.

It appears you needs to a revise your search strategy and search terms to cover all relevant studies.  For example we investigate the role of pregnant women in both quality improvement and assessment activities as customer self-audit. There are many relevant studies using Centering Pregnancy® program to participate women in their own maternity services.  

•  Gholipour K, Tabrizi JS, Jafarabadi MA, Iezadi S, Mardi A. Effects of customer self-audit on the quality of maternity care in Tabriz: A cluster-randomized controlled trial. PloS one. 2018 Oct 11;13(10):e0203255.

•  Gholipour K, Tabrizi JS, Asghari Jafarabadi M, Iezadi S, Farshbaf N, Farzam Rahbar F, Afsharniya F. Customer's self-audit to improve the technical quality of maternity care in Tabriz: a community trial. EMHJ-Eastern Mediterranean Health Journal. 2016;22(5):309-17.)

We look forward to receiving your revised manuscript.

Kind regards,

Kamal Gholipour, PhD

Academic Editor

PLOS ONE

Journal Requirements:

2. Please ensure that you have addressed all items recommended in the PRISMA checklist including identifying the study as a meta-analysis in the title.

3. Thank you for submitting the above manuscript to PLOS ONE. During our internal evaluation of the manuscript, we found significant text overlap between your submission and the following previously published works.

- https://cccrg.cochrane.org/priority-reviews/more-about-our-consumer-engagement-priority-review

- https://doi.org/10.1002/14651858.CD004563.pub2

- https://www.who.int/evidence/sure/PatientSafetyfullreport10052014.pdf?ua=1

We would like to make you aware that copying extracts from previous publications, especially outside the methods section, word-for-word is unacceptable, even for works which you authored. In addition, the reproduction of text from published reports has implications for the copyright that may apply to the publications.

Please revise the manuscript to rephrase the duplicated text, cite your sources, and provide details as to how the current manuscript advances on previous work. Please note that further consideration is dependent on the submission of a manuscript that addresses these concerns about the overlap in text with published work.

Reviewers' comments:

Reviewer's Responses to Questions

**Comments to the Author**

1. Is the manuscript technically sound, and do the data support the conclusions?

Reviewer #1: Yes

Reviewer #2: Yes

2. Has the statistical analysis been performed appropriately and rigorously? 

Reviewer #1: No

Reviewer #2: Yes

3. Have the authors made all data underlying the findings in their manuscript fully available?

Reviewer #1: Yes

Reviewer #2: Yes

4. Is the manuscript presented in an intelligible fashion and written in standard English?

Reviewer #1: Yes

Reviewer #2: Yes

5. Review Comments to the Author

Reviewer #1: Please see attachment. Answer to question 2 above--statistical analysis-- relates to the presentation, and possible synthesis of odds ratios. Please see attached review for more specifics.

xxxxxxxxxxx

Reviewer #2: This is a thorough and careful job, and this work will be of interest to everyone who wishes to promote patient engagement in health services, policy and research. I am recommending acceptance because I do not feel that I need to see it again. I do have a couple of comments.

First, it is surprising that the authors report no specific funding for such a substantial effort. Surely somebody must have been paying salaries for this. Also, the idea that you involved patient advocates and other stakeholders in the work without providing any compensation seems contrary to the spirit of the thing. I think readers would want to know how this was resourced.

It is rather odd to see repeated use of the conditional in the methods. If it turned out that the nature of the available publications did not allow you to use a certain method you can just say that.

On the whole, with the exception of neonatal mortality, the evidence for most of the outcomes seems quite weak. This is not only because of risk of bias, but also because in many cases the investigators looked at multiple outcomes and found evidence for some but not others. The positive findings often pertained to intermediate outcomes and not to more downstream outcomes, as well. While you do generally acknowledge the need for more evidence and you are restrained in endorsing conclusions, I still have an overall feeling that you might be overselling the findings just a bit. Maybe that's just me. Anyway I appreciated this and I think it's a worthy contribution.

6. PLOS authors have the option to publish the peer review history of their article (what does this mean?). If published, this will include your full peer review and any attached files.

Reviewer #1: No

Reviewer #2: **Yes: **

---

## [Author Response · Author response to Decision Letter 0]

22 Jul 2021

Please refer to the attached 'Response to reviewers' document (July 21 updated) which contains detailed responses to each feedback item.

---

## [Editor Report · Decision Letter 1]

13 Dec 2021

Consumer engagement in health care policy, research and services: A systematic review and meta-analysis of methods and effects.

PONE-D-20-36087R1

Dear Dr. Hillier,

We’re pleased to inform you that your manuscript has been judged scientifically suitable for publication and will be formally accepted for publication once it meets all outstanding technical requirements.

Kind regards,

Kamal Gholipour, PhD

Academic Editor

PLOS ONE
---

## [Editor Report · Acceptance letter]

28 Dec 2021

PONE-D-20-36087R1 

Consumer engagement in health care policy, research and services: A systematic review and meta-analysis of methods and effects. 

Dear Dr. Hillier:

I'm pleased to inform you that your manuscript has been deemed suitable for publication in PLOS ONE. Congratulations! Your manuscript is now with our production department. 

Kind regards, 

on behalf of

Dr. Kamal Gholipour 

Academic Editor

PLOS ONE